# A novel clustering approach to bipartite investor-startup networks

**Théophile Carniel** [1,2]*, **José Halloy**[2], **Jean-Michel Dalle**[1,3]

**1** Agoranov, Paris, France, **2** Université Paris Cité, CNRS, LIED UMR 8236, Paris, France, **3** Sorbonne Université, Paris, France

* tc@agoranov.com

## Abstract

We propose a novel similarity-based clustering approach to venture capital investors that takes as input the bipartite graph of funding interactions between investors and startups and returns clusterings of investors built upon 5 characteristic dimensions. We first validate that investors are clustered in a meaningful manner and present methods of visualizing cluster characteristics. We further analyze the temporal dynamics at the cluster level and observe a meaningful second-order evolution of the sectoral investment trends. Finally, and surprisingly, we report that clusters appear stable even when running the clustering algorithm with all but one of the 5 characteristic dimensions, for instance observing geography-focused clusters without taking into account the geographical dimension or sector-focused clusters without taking into account the sectoral dimension, suggesting the presence of significant underlying complex investment patterns.

**Data Availability Statement:** Data cannot be shared publicly because of third party ownership. The data underlying the results presented in the study are available through the Crunchbase Academic Research Access Program (https://about.crunchbase.com/partners/academic-

## Introduction

Within the active field of entrepreneurship research [1], quantitative analyses of the structural properties of investor-startup interactions have been conducted so far on a simplified version of the investor-startup network, namely, on the network of investor-investor relationships, through the construction of syndication networks where two investors are linked if they either invested jointly in a startup or have a common startup in their portfolios [2–4]. The venture capital network, however, actually consists in investor nodes interacting with startup nodes through funding events that occur relatively sparsely and according to a sequence of so-called stages (Seed, Series A, Series B, Series C, etc.). As a consequence, investor-startup interactions could be and should be associated with a temporal bipartite network structure, of which the previously mentioned syndication networks are, in reality, one-mode projections, with valuable structural information being lost in this folding process [5]. These limitations are typically manifest when trying to address and account for the important and structural heterogeneity between investors: startup investors have marked differences, with respect to sectoral specialization, to the average amounts invested (from hundreds of thousands of dollars to hundreds of millions), or else to their geographical focus, to name but a few relevant dimensions. Ignoring this heterogeneity or failing to address it appropriately results in biased, if not misleading,

research-access/) or by subscribing to the Crunchbase API (https://data.crunchbase.com/docs/using-the-api).

**Funding:** The authors received no specific funding for this work.

**Competing interests:** The authors have declared that no competing interests exist.

conclusions, and certainly makes the observation and characterization of larger-scale collective phenomena with respect to entrepreneurial ecosystems and of their temporal dynamics an impossible task. Community detection algorithms [6, 7] have been applied to traditional syndication networks but have either failed to incorporate explicit information about investment stages [8], which typically results in overestimating actors who invest early in startups and are therefore linked to numerous subsequent investors according to syndication links, or have relied on a semi-supervised approach [9] that relies on ex ante and partly subjective and/or largely unavailable segmentation of investors, or else have been structurally limited by the definition of the networks studied: [10], using a modularity-based community detection algorithm, identifies communities of investors based on their interactions, but cannot do so based on their similarity and therefore are unable to address the heterogeneity of structural investors. Syndication networks, as one-mode projections, cannot capture the complex and multi-layered interactions characteristic of bipartite venture networks, and therefore relevant aspects of entrepreneurial ecosystems are lost.

More recent methods such as multi-view data clustering [11–13] are promising, but are not able to deal with our specific constraints: our data is fundamentally bipartite, with each of the views containing different types of data (numerical vs. categorical vs. logarithmic) that are either node-based or edge-based. Specific clustering algorithms incorporating domain-specific knowledge to cluster similar investors through their position and representation along the various axes of the complex bipartite multilayer multigraph are thus necessary in order to study investment dynamics in the investor-startup network.

New analytical tools are required to take advantage of the distinctive structure of these networks and to extract more information, associated with more complete datasets that would allow to build both sides of the bipartite networks and the interactions between them. Fortunately, the use of databases giving both large-scope and in-depth data on investor and startup companies and on their interactions is now rapidly becoming standard [14] while, following notably the ecological literature, methods for bipartite graph analysis have recently become more and more developed and accessible [15]. In this context where both tools and materials have become available, we initiate in this article an enriched analysis of interactions in entrepreneurial networks and ecosystems, with a direct look at the funding events rather than at the syndication shadow they project.

## Objectives

We propose a novel, unsupervised investor clustering approach for entrepreneurial investors that mitigates some of the difficulties described earlier. It was developed both as a direct tool to probe and characterize the typology of actors in venture capital ecosystems and as a methodological building block with respect to the quantitative analysis of the dynamics of entrepreneurial ecosystems. Our method is based on an unsupervised community detection algorithm using a Hellinger-based similarity measure, computed over all pairs of investors, and accounting for 5 well-defined characteristic dimensions to describe investors. As a consequence, the similarity between investors is easily quantifiable and interpretable, compared to traditional clustering method based on machine learning techniques—and although significant progress has been made in terms of interpretability [16]. The similarity graph pruning threshold is the only parameter, and the number of outputted classes is freely determined by the clustering algorithm and is not constrained. As it happens, this method also allows for a controlled modification of the clustering parameters and features, which results in the identification of unexpected community-level patterns that help better understand the dynamics of the different classes of investors.

## Materials and methods

### Dataset

The dataset used for this study was extracted through the Crunchbase API on October 7th, 2020. It contains information on 1 156 085 startups (name, creation date, headquarter location, sectors of activity), 348 020 funding events (target startup, date, investors involved, amount, investment stage), 159 585 investors (name, creation date, investor type, investor location) and 1 067 089 individuals (name, past and current professional experiences, level and sectors of education, company board memberships and advisory roles). We removed the *Software* sector from all startups' sectors of activity as this tag is overly represented (occurs in roughly 25% of startups, almost twice as frequent as the second most frequent tag) and is relatively non-descriptive.

### Investor-startup network

We create a temporal bipartite multigraph where top nodes are the investors, bottom nodes are the startups and edges correspond to funding events between the investor and the startup (see Fig 1 for a schematic representation of the graph). As an investor can fund a startup at several points in time, two nodes can be linked through several temporal edges. We removed nodes for which the geographical information was not available and edges where the financing event was not an investment event (grants, debt financing, etc.), and afterwards removed isolated nodes as they do not take part in the network interactions studied. This process resulted in a network with 65 653 top nodes, 95 329 bottom nodes and 392 204 edges linking these two sets.

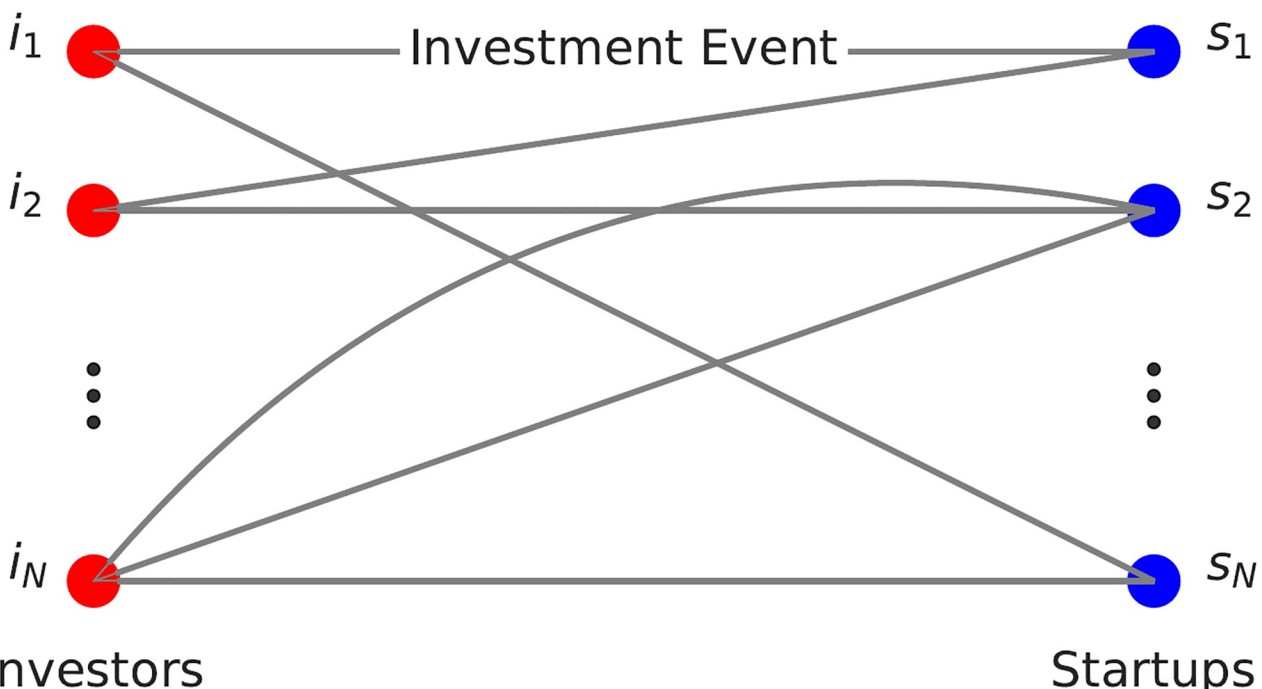

**Fig 1. Schematic representation of the investor-startup multigraph.** The red nodes on the left represent investor nodes, the blue nodes on the right represent startup nodes. The edges between investor node *i* and startup node *s* represent a funding interaction where investor *i* invested in startup *s* at a given time. As an investor can invest in a startup several times, multiple edges can connect two given nodes as shown on the figure.

## Hellinger distance and investor similarity

The Hellinger distance $h$ [17] and the associated similarity $\theta$ between two normalized discrete probability distributions $P$ and $Q$ are defined as:

$$h(P, Q) = \frac{1}{\sqrt{2}} \left\| \sqrt{P} - \sqrt{Q} \right\|_2 \tag{1}$$

$$\theta(P, Q) = 1 - h(P, Q) \tag{2}$$

where $\|.\|_2$ is the Euclidean (or L2) norm [18] and $\sqrt{P}$ is the vector with elements the square root of the elements of $P$. By definition, $0 \leq h(P, Q) \leq 1$ and thus $0 \leq \theta(P, Q) \leq 1$ with $\theta = 0$ corresponding to minimal similarity (maximal distance) and $\theta = 1$ to maximal similarity (minimal distance) between two distributions. The Hellinger distance is used as the probability distributions are low-dimensional and it has been shown to be more suitable than Minkowski distances for probability vector comparisons [19–21].

The similarity $\Theta$ between two investors $\vec{i_a}$ and $\vec{i_b}$ is then defined as follows:

$$\Theta(\vec{i_a}, \vec{i_b}) = \left| \prod_{k=1}^{k=n} \theta(i_a^k, i_b^k) \right|^{1/n} \tag{3}$$

where $i_a^k$ is the distribution characterizing investor $a$ along the $k$-th dimension and $n$ the total number of dimensions characterizing an investor.

## Investor characterization

We characterize investors along $n = 5$ dimensions related to their investments in startups, each of which being associated with a frequency distribution, chosen in order to collectively exhaustively describe investment portfolios and therefore to allow to accurately characterize investors. Within the bipartite graph, these dimensions depend both on **edges** linking an investor to startups (for instance the date of the investment, as several different temporal edges can link an investor and a startup) or on the **startup nodes** (e.g. the geographical location of an investment made by investor *i* targeting startup *s* will be the geographical location of startup *s*). These characteristic dimensions can be measured for all investors, are public enough so that the information is available for most transactions and are linked to common descriptors used by practitioners of the domain to characterize investors (for instance *early-stage* vs. *late-stage* [22], *domestic* vs. *international* [23], *specialized* vs. *generalist* [24], *historical* vs. *emergent* [25]).

- **Temporal investment distribution:** the frequency of investments per year of the investor (Fig 2). This is an edge attribute.

- **Geographical investment distribution:** the frequency of investments of the investor in each country (an investor invests in a country if the target startup's headquarters are located in the country) (Fig 3). This is a startup node attribute.

- **Sectoral investment distribution:** the frequency of investments of the investor in each sector of activity (an investor counts as investing in a sector if the target startup of the investment is labeled in this sector) (Fig 4). This is a startup node attribute.

- **Stage investment distribution:** the frequency of investments of the investor in each stage of the venture capital cycle (Fig 5). This is an edge attribute.

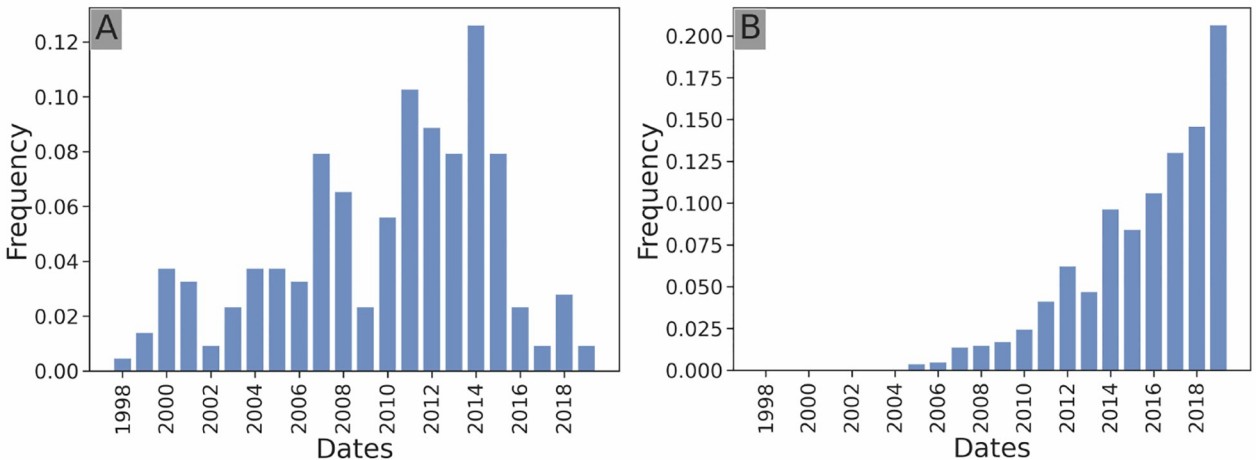

**Fig 2. Temporal investment distribution.** Temporal investment distribution of *Softbank Capital* (**A**), a telecom-focused US-based venture capitalist that stopped its activity in 2017, and of *Y Combinator* (**B**), a US-based startup accelerator founded in 2005. The two temporal patterns of actvitity are quite different between the two structures, as Softbank Capital stops investing near the end of the period whereas Y Combinator's activity steadily grows throughout the whole period.

- **Amount investment distribution:** log-binned distribution of the funding amounts of all investments of the investor in USD (Fig 6). Logarithmic binning was used because the amounts of start-up financing rounds follow a power-law type distribution [26]. This is an edge attribute.

## Self-difference index

For each community $g$ and each year $t$ in the period of study, the set of the top $p$ sectors $k_t^g = \{m_1, m_2, ..., m_p\}$ in terms of number of investment is computed. The self-difference index $d \in$

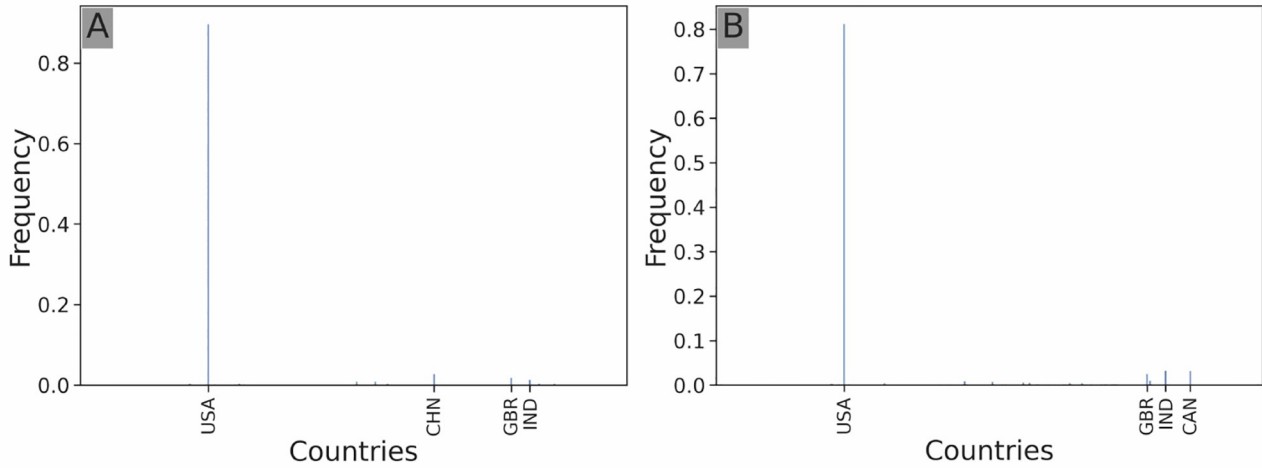

**Fig 3. Geographical investment distribution.** Geographical investment distribution of *Softbank Capital* (**A**), and *Y Combinator* (**B**). Only the top 4 target countries in terms of frequency of investment are labeled. Both structures heavily target US-based ventures.

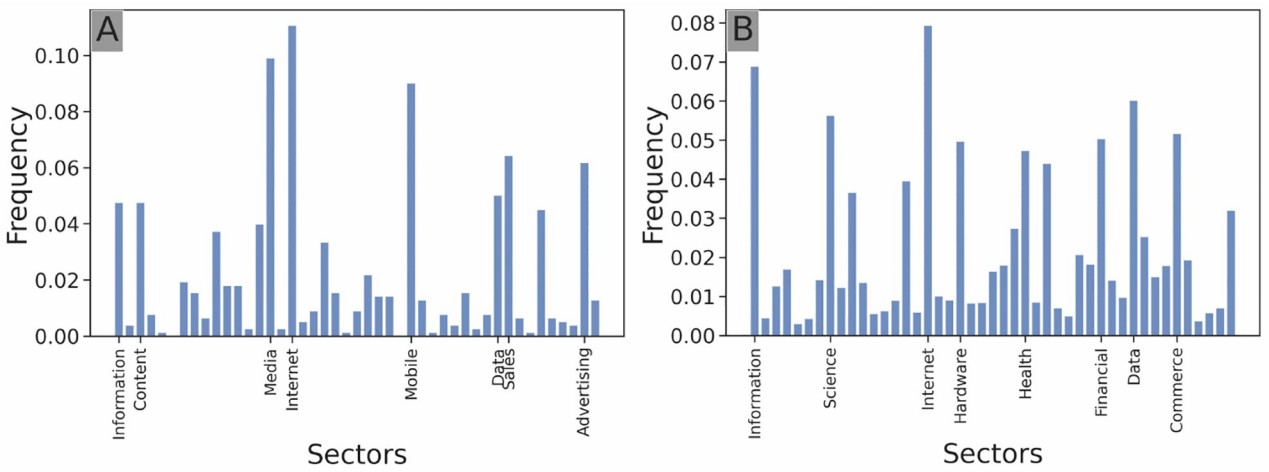

**Fig 4. Sectoral investment distribution.** Sectoral investment distribution of *Softbank Capital* (**A**) and *Y Combinator* (**B**). Only the top 8 sectors of investment are labeled. Softbank Capital shows a strong focus on IT-related ventures whereas Y Combinator shows a wider sectoral breadth.

$[0, 1]$ between years $t_1$ and $t_2$ for community $g$ is defined as follows:

$$d(k_{t_1}^g, k_{t_2}^g) = \frac{k_{t_1}^g \Delta k_{t_2}^g}{2 \min(P - p, p)} \tag{4}$$

where $\Delta$ is the symmetric difference between both sets and $P$ is the total number of sectors. This self-difference index ranges from 0 (identical sets) to 1 (no overlap between the top $p$ sectors of investment at year $t_1$ and the top $p$ sectors of investment at year $t_2$). As there is a natural inflation in terms of number of investment rounds due to an increase in venture capital activity during the latter part of the period of study, the index takes into account the ordering of the sectors in terms of number of investments rather than the raw number of investments.

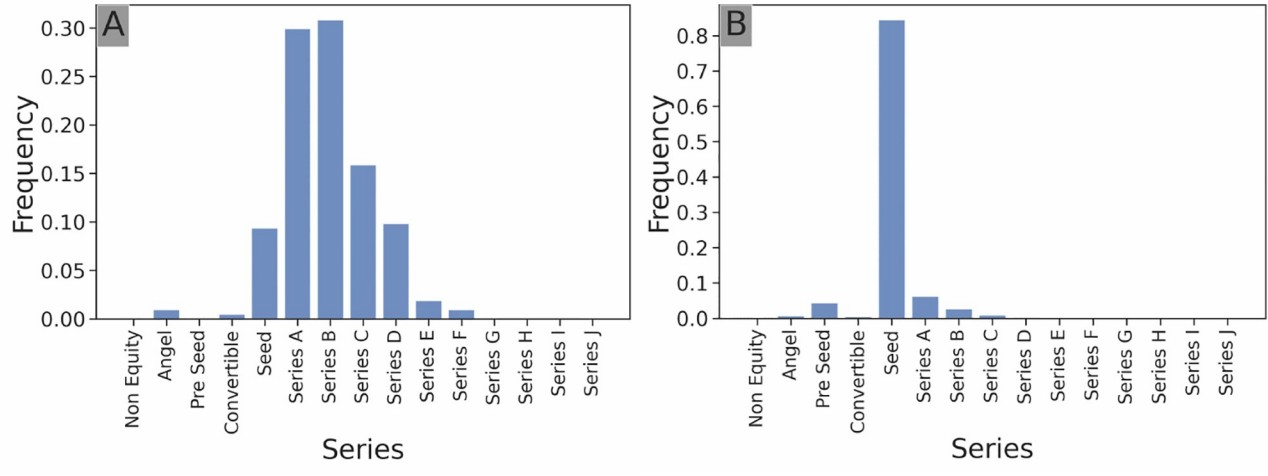

**Fig 5. Stage investment distribution.** Stage investment distribution of *Softbank Capital* (**A**) and *Y Combinator* (**B**). Softbank Capital shows a strong focus in late-stage investment (most of its investments are in Series B or later) whereas Y Combinator shows a very strong early-stage specialization (over 80% of its investments in Seed stage).

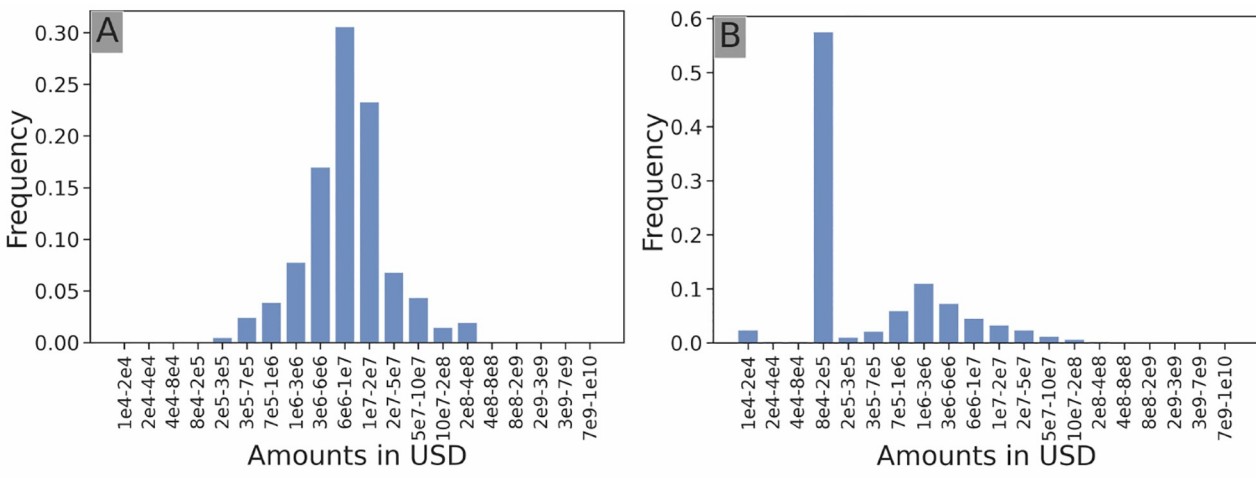

**Fig 6. Amount investment distribution.** Amount investment distribution of *Softbank Capital* (**A**) and *Y Combinator* (**B**). In line with Fig 5, we see that Softbank Capital invests relatively high amounts (peak frequency of investment between 6 million USD and 10 million USD) whereas Y Combinator invests smaller amounts in a very systematic manner (peak frequency of investment between 80 000 USD and 200 000 USD). This is in line with the accelerator model where accelerators invest a set amount in all ventures they decide to support. Furthermore, Y Combinator has also developed funds such as Y Combinator Continuity dedicated to investing in its alumni companies after their initial investment. This can be seen in the small bump in the funding amount distribution between 700 000 USD and 10 million USD.

## Results

### Investor communities

**Clustering.** We reduce the set of top nodes (investors) worldwide to top nodes with degree $d \geq 60$ investments throughout the 1998–2019 period (a low number for a professional investor over this time frame) to ensure a sufficient number of observations for each dimension characterizing an investor. Note that the same clustering results hold for a graph reduced to investors with $d \geq 100$ or more investments. This procedure results in 1014 investor nodes in the final graph with 159 353 edges connecting them to startup nodes, isolate nodes being removed (see previous section). We compute the pairwise similarity $\Theta$ as defined in Eq 3 between all investors in our sample and then define a complete weighted similarity graph with investors as nodes and the similarity between two investors as edge weights. We prune the graph by retaining for each investor the 1% edges with the highest similarity. We then run the *best_partition* community detection algorithm from the Python *community* package [27] resulting in an investor clustering with 11 different communities.

For each of the communities, a theoretical *representative investor* defined as the barycenter of the communities' investors in the 5-dimensional probability space is computed: in each dimension, the distribution of the representative investor of a given community is the average of the distributions of all investors in the community. This representative investor allows for a compact visualization and understanding of each community, yielding some relevant understanding as to how the communities are formed. Fig 7 for instance shows the representative investor for community **A6** and shows that investors in community **A6** have an obvious China-focused geographical bias since over 84% of the cluster's investments target China-based startups. As another example, Fig **S11** in the S1 File shows a similar sectoral focus on Health Care-related investments in community **A7**, with around 27%, 30% and 26% of investments in *Science and Engineering*, *Health Care* and *Biotechnology* respectively.

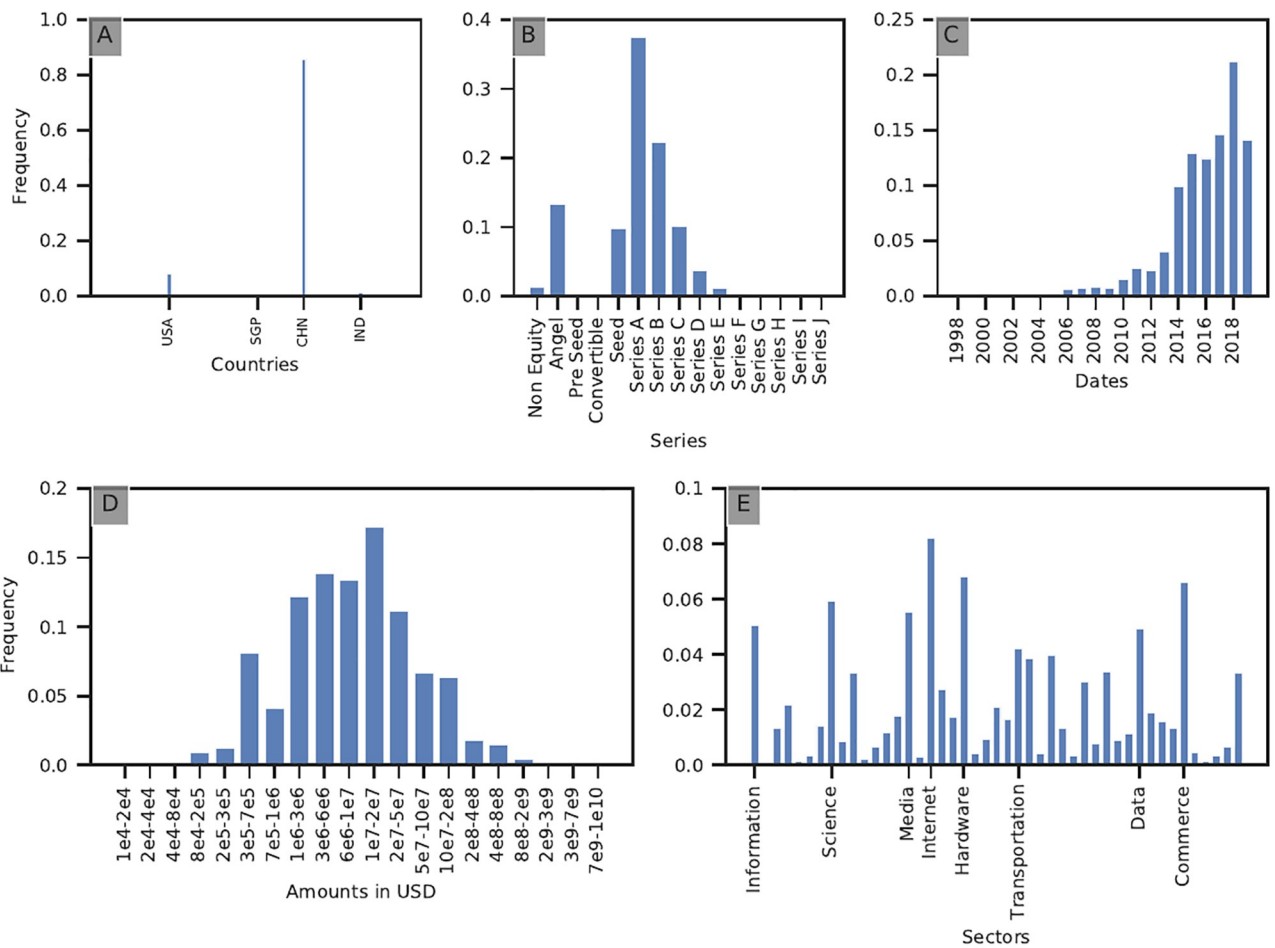

**Fig 7. Representative investor of community A6.** Community **A6** appears comprised of investors targeting China-based ventures during the second half of the 2010s with no clear sectoral specialization. Panel **A** shows the representative geographical investment distribution of community **A6**, panel **B** the distribution of the series of investment, panel **C** the temporal distribution of investments, panel **D** the distribution of the amounts of investment and panel **E** shows the sectoral distribution of investment.

Fig 8 shows the similarity graph pruned as described previously without (left) and with (right) the results of the clustering superimposed on the individual nodes. In light of these observations, we further characterize each of the resulting communities as described in column **A** of Table 1 by analyzing the representative investors of each of the 11 communities, which can be found in the S1 File (Figs **S4**-**S14**)), and referring also to the identity of individual investors in the clusters (see Table 2 for a sample of individuals from each cluster). We observe that each community corresponds to a strong and specific pattern: a specific geographical area of investment, a specific sector of investment, investing at specific startup development stages, or displaying a specific temporal pattern notably in relation to the 2008 financial crisis i.e. grouping investors that were either active throughout the whole period, or that belonged to older or newer generations of investors typically active either before or after the 2008 crisis.

**Temporal evolution patterns.** Based on this investor clustering, Figs 9 and 10 reveal the temporal evolution of two communities in terms of target sectors of investment over the 2010–2019 period. Community **A0**, composed of general investors active over the whole period studied, typically shows a relatively slow evolution in terms of sectoral trends, with a

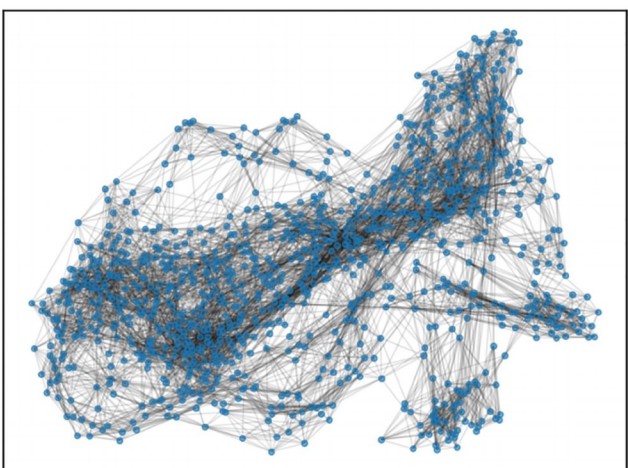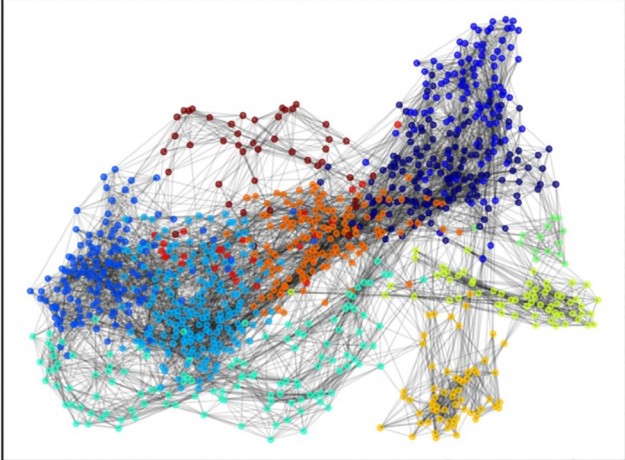

**Fig 8. Similarity graph and community assignment.** Pruned similarity graph without (left) and with (right) community assignment of the nodes as characterized in column **A** of Table 1. The neon yellow community corresponds to China-focused venture capital firms (**A6**), the dark red community to India and Japan-focused venture capital firms(**A10**), the gold community to Health Care specialists (**A7**), the blue community (far left) to accelerators (**A2**).

gradual shift (Fig 9) in preferred sectors of investment towards so-called *deeptech* sectors (shift from sectors such as *Media and Entertainement*, *Mobile* towards sectors such as *Science and Engineering*, *Health Care*). Community **A7**, composed of health-care focused investors, shows a very strong dominance of Health Care-related sectors throughout the whole period (Fig 10A), but where the top 10 sectors have significantly evolved over the 10-year period of study (Fig 10B). A closer look at the non-health related sectors reveals a clear shift from *Manufacturing* and *Hardware*-related investments towards *Data Science and Analytics* and *Artificial Intelligence*-related investments, in line with the widespread adoption of these technologies in Health Care-related sectors during recent years [30].

## Clustering factor analysis highlights underlying investment patterns

Since the 5 characteristic dimensions are based on domain knowledge, we ran the clustering algorithm 5 additional times, each time using only 4 of the 5 dimensions previously defined, computing the representative investors of all communities for each of these alternative clusterings in order to understand the characteristics of the new communities. Fig **S21** in S1 File shows the representative investor of community **B6** resulting from a clustering without the geographical investment dimension. Surprisingly, the community shows a strong focus on the Chinese startup market, with around 80% of all investments targeting China-based startups although the geographical dimension was not taken into account, therefore suggesting the existence of an underlying structure: the existence of an investment pattern according to the 4 other investment dimensions that is actually characteristic of investors investing mostly in China. Similarly, Fig **S32** in S1 File shows the representative investor of community **C7** resulting from a clustering without the sectoral dimension, but shows a community strongly focused on Health Care startups (around 17%, 18% and 15% of investments in *Science and Engineering*, *Health Care* and *Biotechnology* respectively) not unlike the community shown in Fig **S11** in S1 File, even though sectors were not taken into account in this clustering.

Following these observations, we systematically investigate the bivariate distributions for all pairwise combinations for each alternative clustering, with the discrete bivariate distribution $f$

**Table 1. Descriptive table of the communities for the different clusterings.**

| Community | Complete Clustering (A) | Clustering Without Countries (B) | Clustering Without Sectors (C) |
|---|---|---|---|
| 0 | General investors active whole period | General investors active whole period<br>Similarity with community A0: 0.931 | General investors active whole period<br>Similarity with community A0: 0.956 |
| 1 | General investors active pre-2008 crisis | General investors active pre-2008 crisis<br>Similarity with community A1: 0.960 | General investors active pre-2008 crisis<br>Similarity with community A1: 0.96 |
| 2 | Accelerators [28, 29] | Accelerators and incubators<br>Similarity with community A2: 0.92 | Accelerators<br>Similarity with community A2: 0.915 |
| 3 | Early-stage investors post-2008 crisis | Early-stage investors low amounts post-2014<br>Similarity with community A3: 0.857 | Early-stage investors post-2008 crisis<br>Similarity with community A3: 0.935 |
| 4 | EU-focused investors | Early-stage investors low amounts post-2008 crisis<br>Similarity with community A3: 0.887 | Canada-focused investors<br>Similarity with community A9: 0.956 |
| 5 | Late-stage investors | Late-stage investors<br>Similarity with community A5: 0.882 | General investors active post-2008 crisis<br>Similarity with community A8: 0.964 |
| 6 | China-focused investors | China-focused investors<br>Similarity with community A6: 0.954 | EU-focused investors<br>Similarity with community A4: 0.870 |
| 7 | Health Care-focused investors | Health Care-focused investors<br>Similarity with community A7: 0.988 | Health Care-focused investors<br>Similarity with community A7: 0.813 |
| 8 | General investors active post-2008 crisis | General investors active post-2008 crisis<br>Similarity with community A8: 0.877 | China-focused investors<br>Similarity with community A6: 0.989 |
| 9 | Canada-focused investors | "Next-generation" post-2014 general investors<br>Similarity with community A8: 0.874 | Japan and India-focused investors<br>Similarity with community A10: 0.978 |
| 10 | Japan and India-focused investors | | "Next-generation" post-2014 general investors<br>Similarity with community A3: 0.863 |
| 11 | | | |

| Community | Clustering Without Time (D) | Clustering Without Series (E) | Clustering Without Amounts (F) |
|---|---|---|---|
| 0 | General investors active whole period<br>Similarity with community A8: 0.899 | General investors active whole period<br>Similarity with community A0: 0.969 | General investors active whole period<br>Similarity with community A0: 0.956 |
| 1 | Middle-stage investors active whole period<br>Similarity with community A0: 0.881 | Early-stage investors active post-2008 crisis<br>Similarity with community A3: 0.904 | General investors active pre-2008 crisis<br>Similarity with community A1: 0.988 |
| 2 | General investors active post-2008 crisis<br>Similarity with community A8: 0.908 | UK-focused early-stage investors Similarity with community A4: 0.791 | North America-focused incubators<br>Similarity with community A9: 0.805 |
| 3 | North America-focused incubators<br>Similarity with community A9: 0.814 | General investors active pre-2008 crisis<br>Similarity with community A1: 0.976 | Accelerators<br>Similarity with community A2: 0.888 |
| 4 | EU-focused investors<br>Similarity with community A4: 0.885 | EU-focused investors<br>Similarity with community A4: 0.886 | UK-focused early-stage investors<br>Similarity with community A4: 0.791 |

(*Continued*)

**Table 1.** (Continued)

| | | | |
|---|---|---|---|
| 5 | Very early-stage investors active post-2008 crisis (UK and US) Similarity with community A2: 0.868 | "Next-generation" post-2014 general investors Similarity with community A8: 0.899 | EU-focused investors Similarity with community A4: 0.898 |
| 6 | Early-stage investors active post-2008 crisis Similarity with community A3: 0.949 | China-focused investors Similarity with community A6: 0.908 | General investors active post-2008 crisis Similarity with community A8: 0.923 |
| 7 | General investors active pre-2008 crisis Similarity with community A1: 0.933 | Health Care-focused investors Similarity with community A7: 0.969 | Early-stage investors active post-2008 crisis Similarity with community A3: 0.901 |
| 8 | Israel-focused investors Similarity with community A0: 0.829 | Canada-focused investors Similarity with community A9: 0.977 | "Next-generation" post-2014 general investors Similarity with community A3: 0.859 |
| 9 | China-focused investors Similarity with community A6: 0.992 | Accelerators Similarity with community A2: 0.929 | China-focused investors Similarity with community A6: 0.992 |
| 10 | Health Care-focused investors Similarity with community A7: 0.990 | Japan and India-focused investors Similarity with community A10: 0.953 | Health Care-focused investors Similarity with community A7: 0.973 |
| 11 | Japan and India-focused investors Similarity with community A10: 0.973 | | Japan and India-focused investors Similarity with community A10: 0.976 |

Each clustering is denoted by a letter and each community by a number (i.e. community **B4** corresponds to community **4** for the clustering **without the geographical dimension**). The second line in each cell denotes the community from clustering **A** that is most similar and the associated similarity value. The similarity value is computed between the representative investors of said community and all communities of the complete clustering following Eq 3.

of group $g$ at coordinates $(m, n)$ defined as:

$$f_g(m, n, k_1, k_2) = \frac{\sum_{\epsilon=1}^{\epsilon=T} i_\epsilon^{k_1}(m) \ i_\epsilon^{k_2}(n)}{\sum_{v=1}^{v=V}\sum_{w=1}^{w=W}\sum_{\epsilon=1}^{\epsilon=T} i_\epsilon^{k_1}(v) \ i_\epsilon^{k_2}(w)} \qquad (5)$$

where investor distribution $k_1$ has dimension $V$ and $k_2$ has dimension $W$ with group $g$ being comprised of $T$ investors.

**Geographical.** Fig 11 shows the resulting bivariate distribution for all pairs of dimensions for community **B6**, here presented as heatmaps. It shows that **B6** investors take part mostly in series A investments between \$10M and \$20M after 2015, which could correspond to a pattern characteristic of China-focused investors in our sample. For all bivariate distributions shown in Fig 11 (community **B6**) and Fig 12 (community **A6**), both communities display virtually identical behaviors: most likely due to this underlying investment pattern, taking into account the geographical dimension is *not* necessary to characterize this cluster despite its very strong geographical footprint.

**Sectoral.** Similarly, Fig 13 shows the resulting bivariate distribution for all pairs of dimensions for community **C7**. It shows that **C7** investors invest mainly in series B rounds between \$20M and \$50M in North American ventures, which appears to be an investment pattern for investors specialized in Health Care in our sample. Fig 14 shows community **A7** resulting from the complete clustering. Figs 13 and 14 show a strong agreement in terms of *Series* and

**Table 2. Complete clustering: Sample investors from each community.**

| Cluster 0 | Cluster 1 | Cluster 2 | Cluster 3 |
|---|---|---|---|
| CRV | Threshold Ventures | Masschallenge | Marc Cuban |
| Greylock | Venrock | Skydeck Berkeley | Band of Angels |
| Battery Ventures | Sigma Partners | MIT Media Lab | SV Agel |
| RRE Ventures | Fidelity Ventures | 500 Startups | Scott Banister |
| Bain Capital Ventures | H.I.G. Capital | Techstars | Fabrice Grinda |
| GGV Capital | ABS Ventures | Y Combinator | Alexis Ohanian |
| Goldman Sachs | Polaris Partners | Kima Ventures | Betaworks |
| Kleiner Perkins Caufield Byers | Baird Capital | Start-Up Chile | Angelpad |
| Sequoia Capital | Cedar Fund | SOSV | Kickstart Seed Fund |
| Benchmark | Enterprise Partners | Chinaccelerator | Lerer Ventures |
| **Cluster 4** | **Cluster 5** | **Cluster 6** | **Cluster 7** |
| Seedcamp | Tiger Global | IDG Capital Partners | Sofinnova Ventures |
| Amadeus Capital Partners | Temasek | Ceyuan Ventures | Abingworth Management |
| Balderton Capital | KKR | SIG China | Frazier Healthcare Ventures |
| Index Ventures | T. Rowe Price | Shenzhen Capital Group | Sante Ventures |
| Partech | General Atlantic | Sequoia Capital China | SV Life Sciences |
| Alven Capital | Wellington Management | Vertex Ventures China | Orbimed Advisors |
| Xange Private Equity | Coatue | Qingsong Fund | Life Sciences Partners |
| IDInvest Partners | Iconiq Capital | Zhenfund | Oxford Bioscience Partners |
| Bayern Kapital | Google Capital | Baidu | Lilly Ventures |
| Iris Capital | Softbank Vision Fund | Matrix Partners China | Deerfield Management Company |
| **Cluster 8** | **Cluster 9** | **Cluster 10** | |
| Silverton Partners | Celtic House Venture Partners | Mitsubishi UFJ Capital | |
| First Round Capital | BDC Venture Capital | Mizuho Capital | |
| Greycroft | Fonds de Solidarite FTQ | SMBC Venture Capital | |
| Andreessen Horowitz | Inovia Capital | Omidyar Network | |
| Ridge Ventures | Relay Ventures | Sequoia Capital India | |
| GE Ventures | Innovacorp | East Ventures | |
| Foundry Group | Anges Quebec | Mumbai Angels | |
| Miramar Venture Partners | Founderfuel | Innovation Network Corp of Japan | |
| Lux Capital | Venture Alberta | Nissay Capital | |
| IA Ventures | Creative Destruction Lab | Itochu Technology Ventures | |

Ten investors are manually chosen from each community to provide insights about the typology of investors.

*Amounts* of investments but still display slight differences as community **A7** has been active for a longer time than community **C7**. We therefore observe different *generations* of Health Care-focused investors with the newer generations associated with a wider scope of investment in terms of sectors. These new investors tend to invest in Health Care-oriented companies with a stronger IT component in the latter part of the 2010s (see Fig 13), a pattern not found in Fig 14. This suggests that the current shift in Health Care venture funding (linked notably to the use of Artificial Intelligence solutions) could on a global level not be the result of a shift of focus of traditional Health Care-focused investors but rather the outcome of the emergence of a new group of investors in the domain.

**Temporal.**    Again in a similar manner, and analyzing this time the clustering computed without the temporal dimension, Fig S43 in S1 File shows the representative investor of community **D7**, associated with a very specific temporal pattern of investment that appears

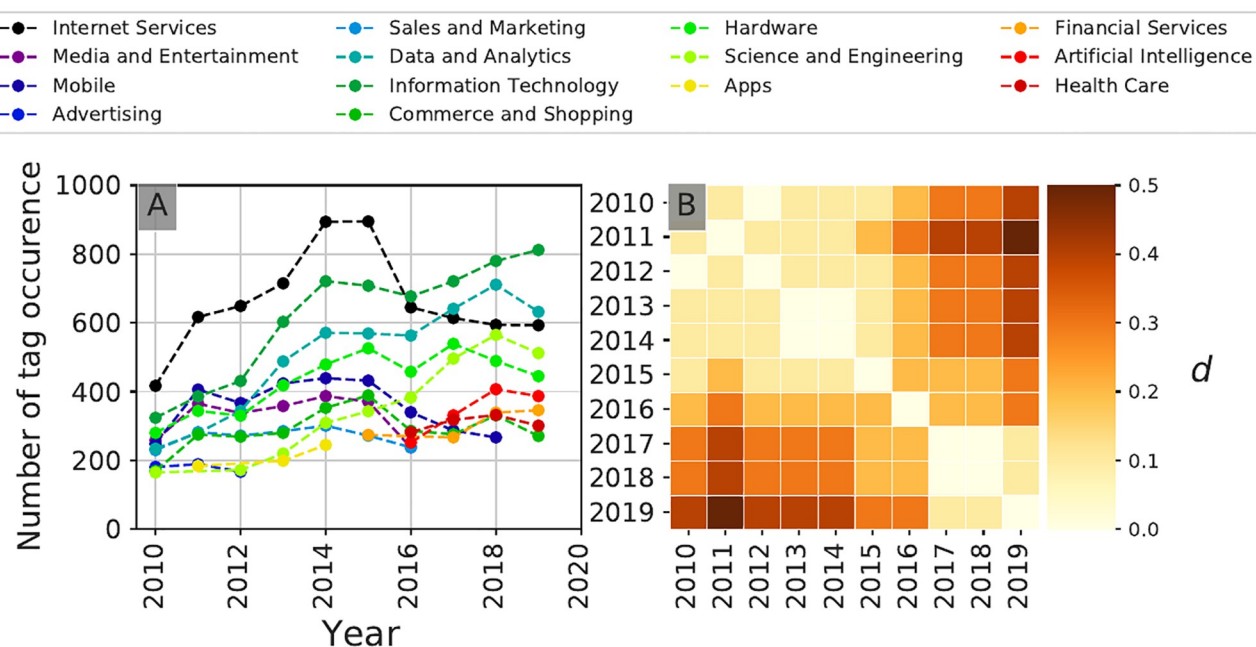

**Fig 9. Temporal evolution of the investment patterns of community A0.** Temporal community investment patterns of the target startups' sectoral tags for each year aggregated at the community level. Community **A0** is comprised of large, historical, rather late-stage focused venture capital firms. Panel **A** shows for each year the ten tags that received the most investments, panel **B** shows the community self-difference index described in Eq 4. We see a gradual but consequent shift in the target industries of community **A0** throughout the period of study as evidenced in panel **B**, notably with the disappearance of relatively low-tech sectors such as the *Mobile*, *Apps* and *Advertising* sectors.

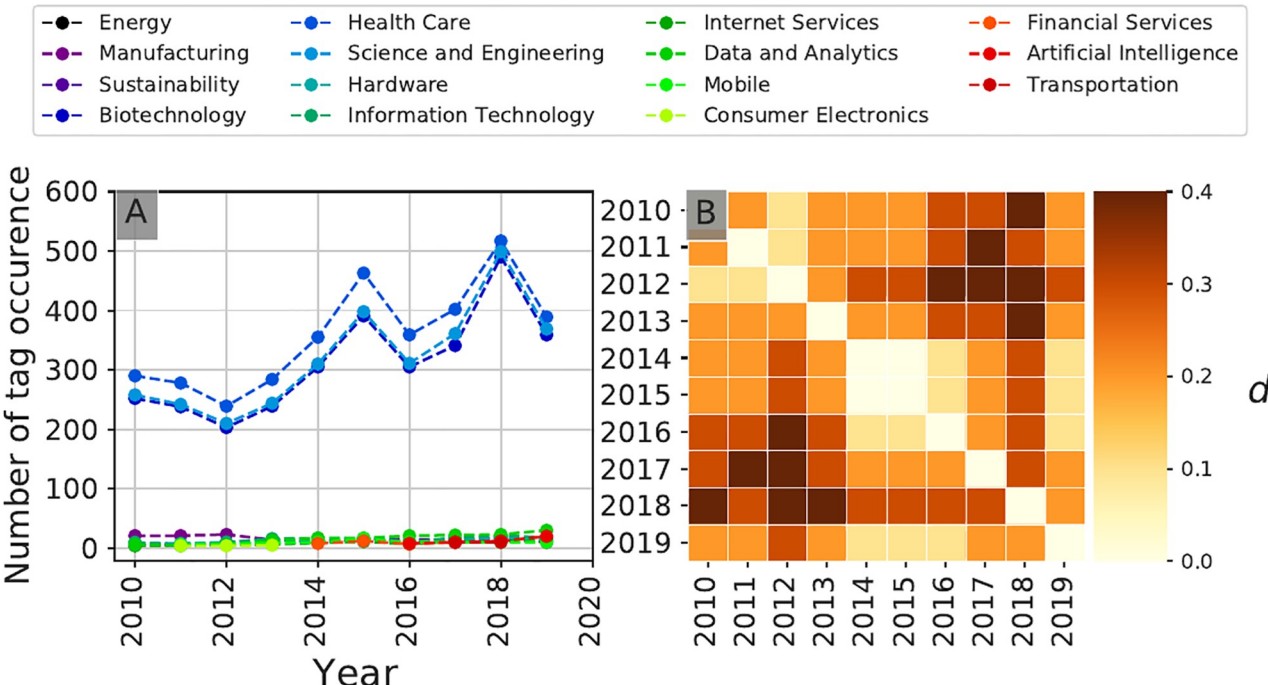

**Fig 10. Temporal evolution of the investment patterns of community A7.** Temporal community investment patterns of the target startups' sectoral tags for each year aggregated at the community level. Community **A7** is comprised of Health Care-specialized venture capitalists. Panel **A** shows for each year the ten tags that received the most investments, panel **B** shows the community self-difference index described in Eq 4, with two markedly different areas of coherence, before and after 2014–2015.

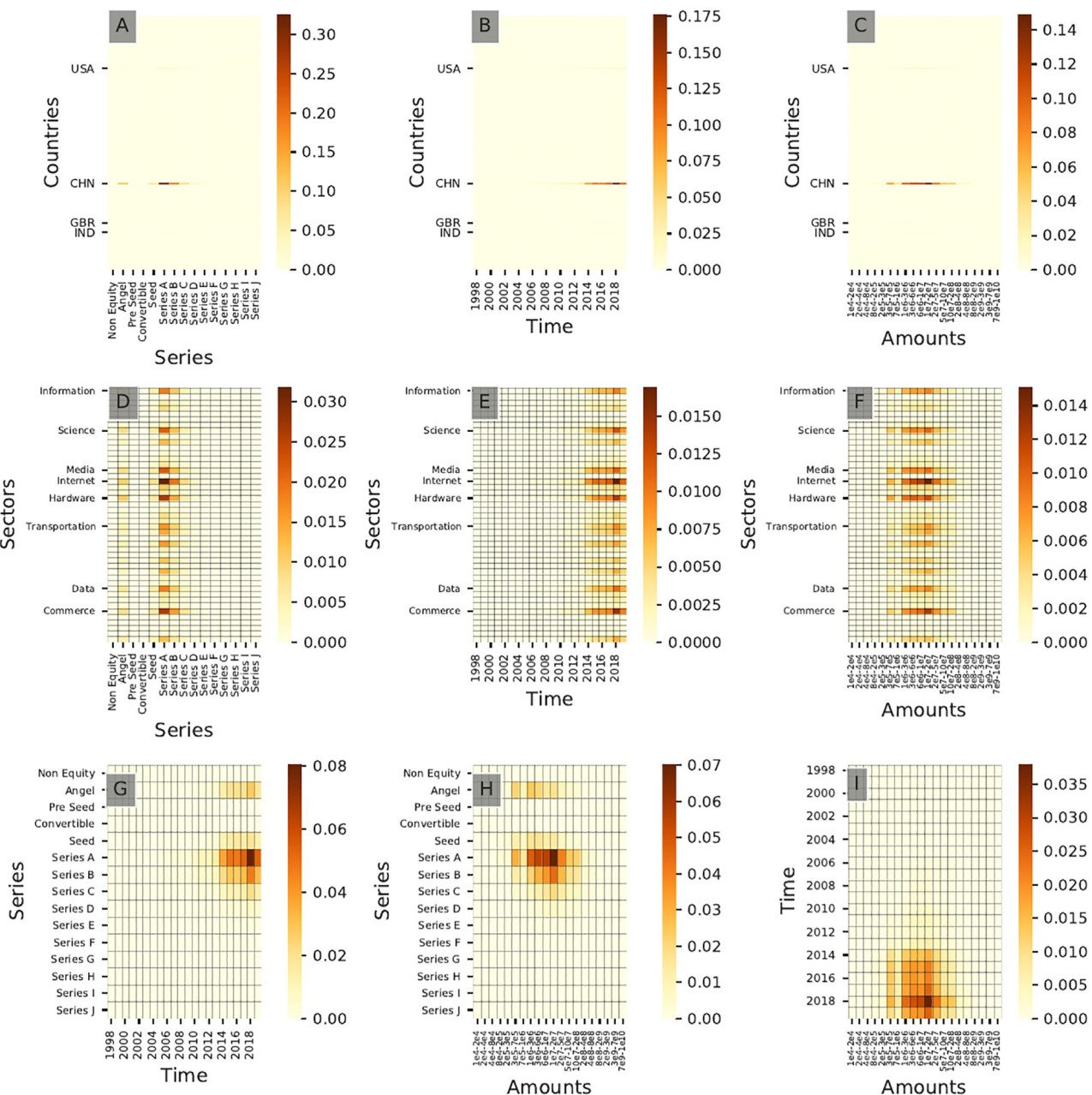

**Fig 11. Cross-interaction heatmaps for community B6.** This community corresponds to China-focused investors. Only the top 8 sectors and the top 4 countries in terms of frequency of investments are labeled for readability purposes.

markedly similar to community **A1** from the complete clustering (see Fig **S5** in S1 File), even though the temporal dimension was excluded in the case of **D7**. This observation therefore again suggests the existence of underlying investment patterns associated with investors. Here, historical, older generation investors appear to have been clustered together independently of their temporal activity, and rather on the basis of a qualitatively specific investment pattern that differs from those of newer generation venture capital firms.

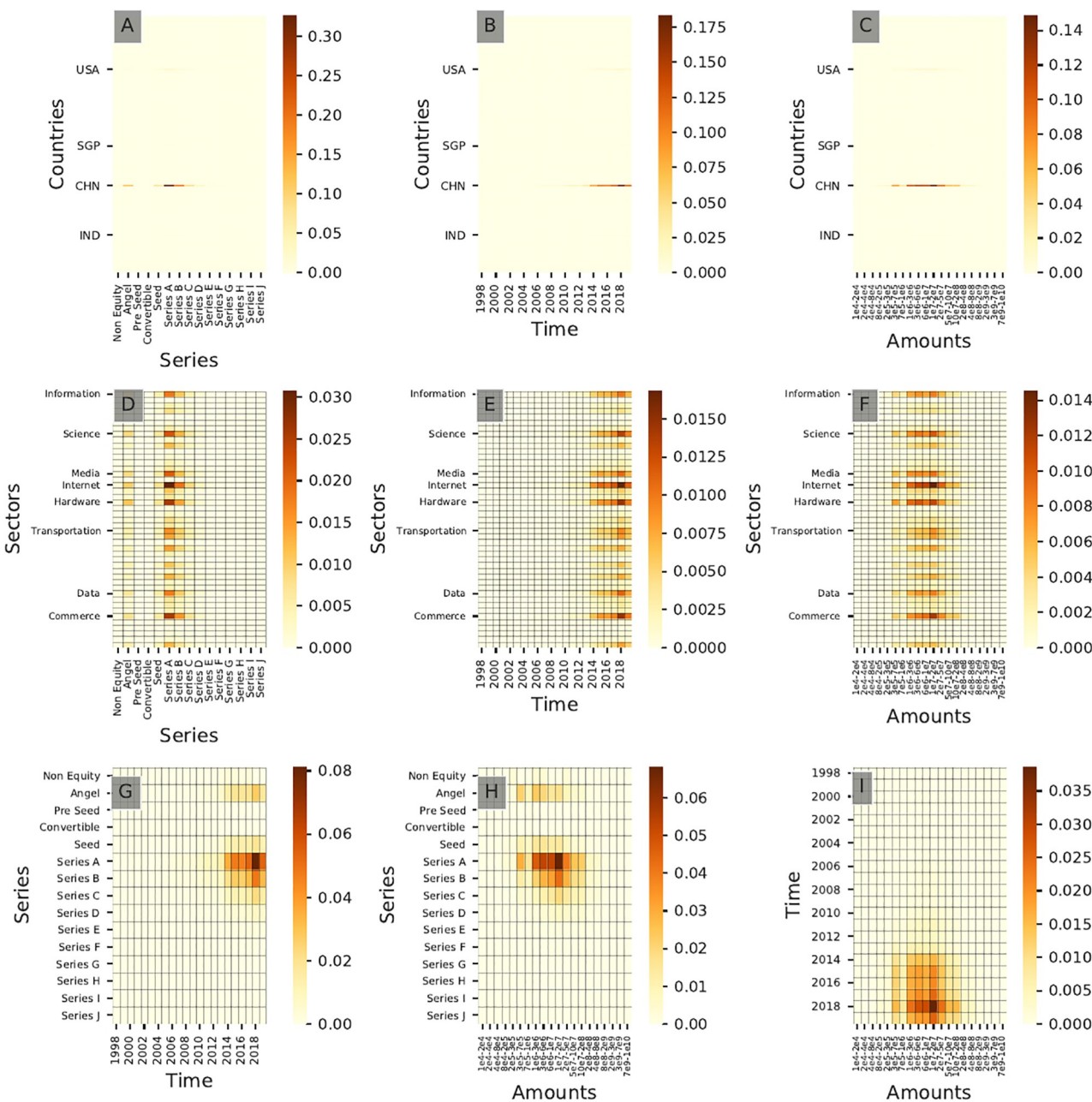

**Fig 12. Cross-interaction heatmaps for community A6.** This community corresponds to China-focused investors.

## Conclusion

In this article, we approached investors through clustering methods in order to help us and fellow researchers make a better sense of the "venture capital community", perhaps in the sense of advocating for the end of their analysis as that of an homogeneous community. We thus described a novel approach to quantitatively group startup investors based only on the characteristics of their investments, as gathered from a bipartite investor-startup network. This clustering approach results in interpretable and homogeneous subgroups of investors with

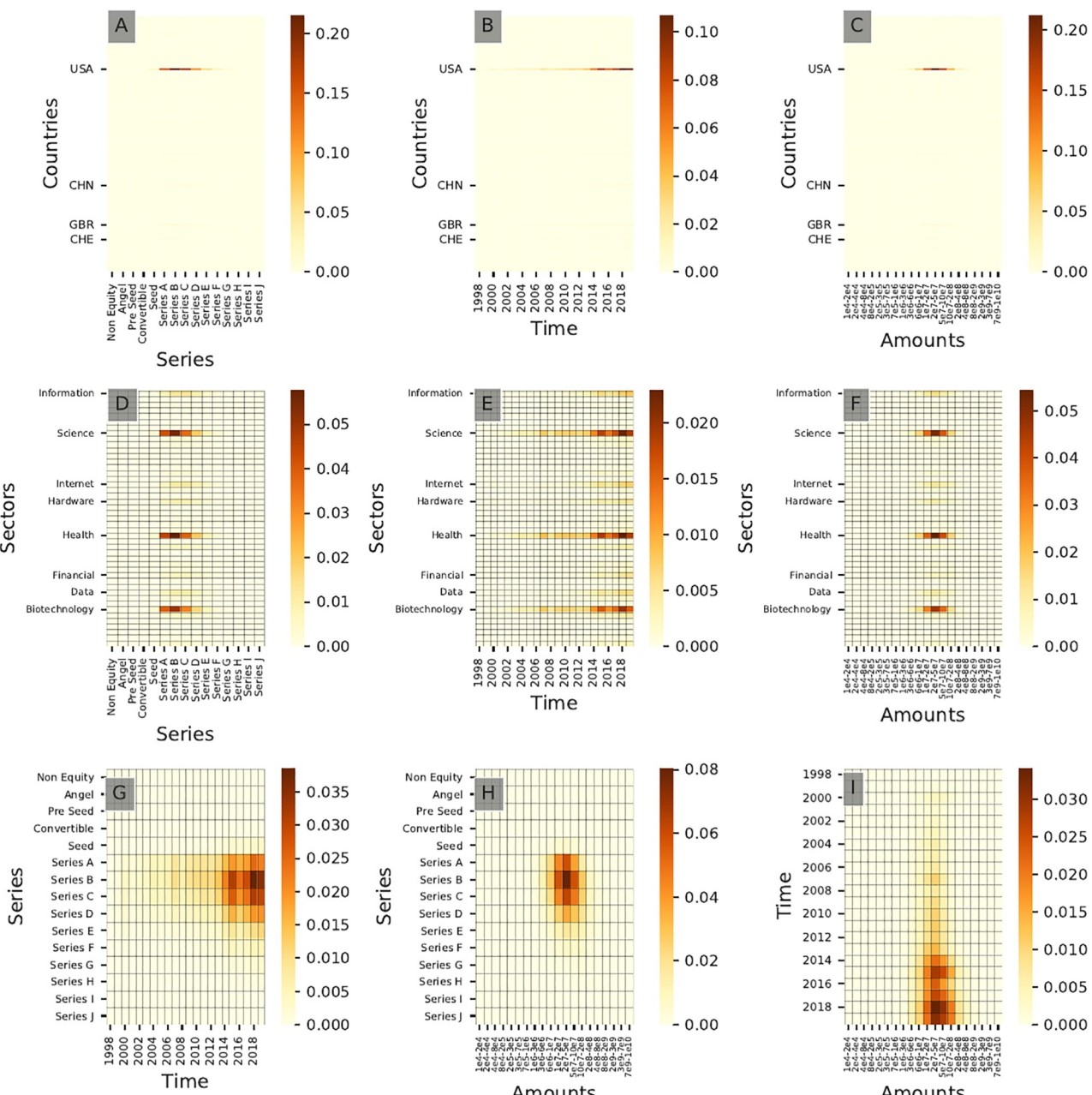

**Fig 13. Cross-interaction heatmaps for community C7.** This community corresponds to a Health Care-focused community of investors. Only the top 8 sectors in terms of total number of investments and the top 4 countries of investment are labeled for readability purposes.

markedly different profiles, which we hope could prove helpful for the community of researchers interested in studying venture capital communities and networks by allowing them to differentiate *among* venture capitalists. In that sense, "the" venture capital community, as often referred to, might actually be composed of several venture capital communities whose investment behaviors and in particular whose co-investment behaviors might considerably differ. As a consequence, we would plead for some of the literature on venture networks to be assessed

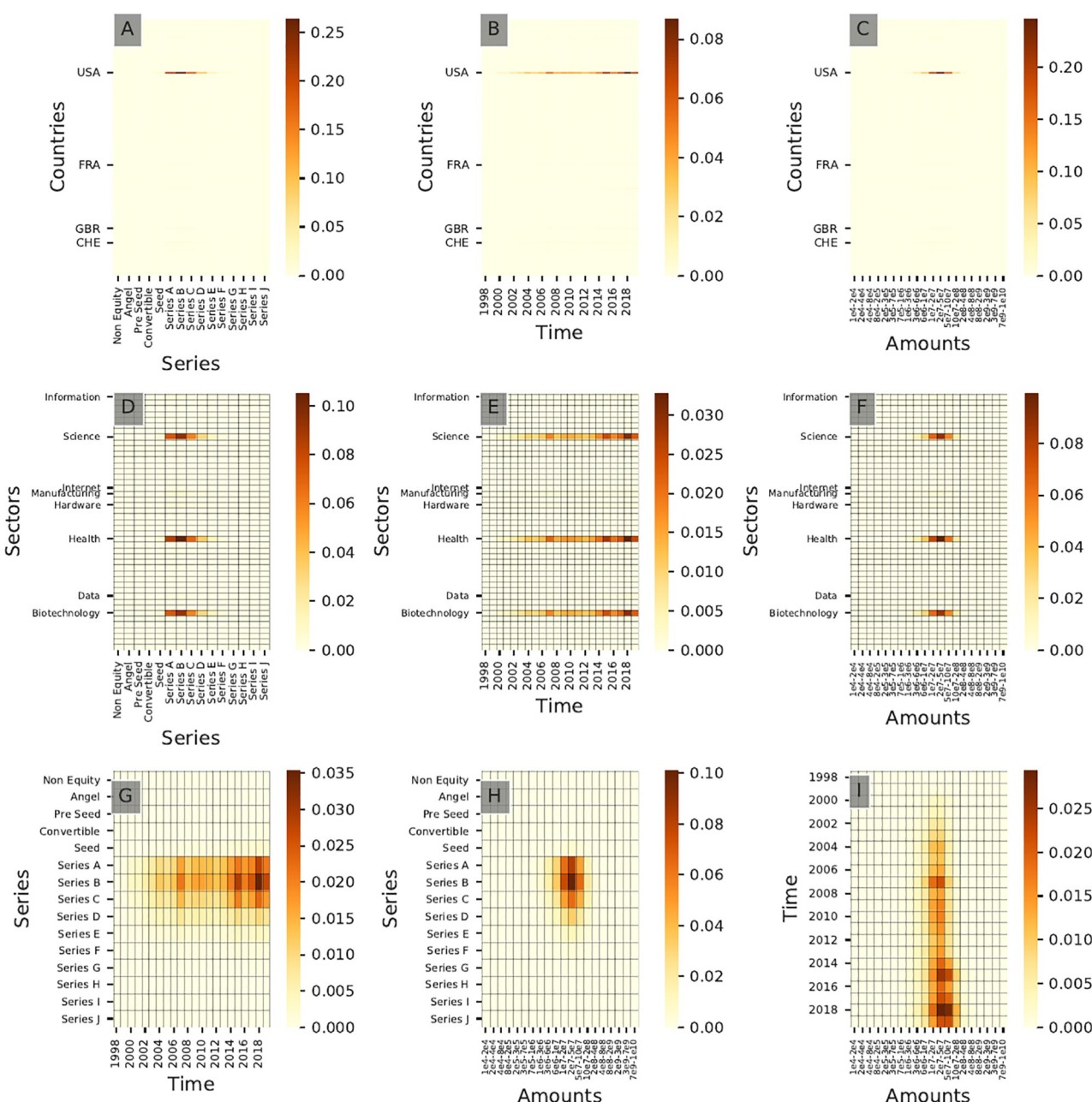

**Fig 14. Cross-interaction heatmaps for community A7.** These distributions correspond to Health Care specialists.

again on each of the venture communities separately, for instance with respect to the relationship between network position and centrality and the profitability of venture investments.

In addition, and by allowing the conditions under which investors are clustered according to our approach to vary, notably by reducing the number of characteristic dimensions taken into account, we were able to observe the presence of relatively surprising underlying and robust investment patterns characteristic of certain clusters of startup investors. For instance, the fact that some investors specialize as Health Care specialists seems to have consequences with respect to their other investment patterns notably in terms of funding amounts or

funding rounds: we did observe a cluster of Healh Care-focused investors even when the sectoral dimension was not accounted for in the clustering. Similarly, the fact that some investors focus on investments in China also results in the existence of patterns with respect to their investment behaviors, once again in terms of funding amounts and funding rounds in particular: we indeed observed a cluster of investors focused on China even when the geography of investments was not taken into account. From a research point of view, these observations raise the issue of whether they would be the result of a behavioral phenomena or rather market outcomes. More broadly, the existence of such underlying patterns could also result in modifying how financial actors directly interpret and evaluate opportunities, compared then to such benchmarks.

Furthermore, similar underlying investment patterns were also observed to characterize different generations of investors, notably in relation to the 2008 financial crisis. We notably observed a cluster of investors mostly active before the 2008 crisis even when the temporal distribution of their investments was not taken into account. In our sample, this observation is particularly striking with respect to the aforementioned crisis, but we also observed preliminary evidence of a similar phenomenon in the case of Health Care focused investors with 2014 as a breaking point, which we can relate to the significant increase in startup investment activity that occurred around that date. Altogether, and adding also that the cluster of so-called accelerators (**A2**) also corresponds to a completely new "species" of investors that appeared in the late 2000s, these preliminary observations might suggest a mechanism that would evoke the notion of *speciation* in ecology: whenever the "financial environment" would change, newer "species" of investors could appear in an evolutionary way, by seizing the newer opportunities offered by the new environment, while existing investors might either adapt or stay locked in their previous patterns even though these patterns might eventually not represent an adaptive advantage in a new financial environment. Rather than simply suggesting an evolutionary perspective, these observations could also shed more light on the determinants of success for so-called "Limited Partners" [31], i.e. investors in venture capital funds, by potentially providing a supplementary explanation of why returns would differ systematically across limited partners [32]. They could also provide limited partners and other actors in the finance community themselves with a new understanding of the dynamics of innovation in the venture capital market.

## Supporting information

**S1 File.**
(PDF)

## Author Contributions

**Conceptualization:** Théophile Carniel, José Halloy, Jean-Michel Dalle.

**Data curation:** Théophile Carniel.

**Formal analysis:** Théophile Carniel, José Halloy, Jean-Michel Dalle.

**Investigation:** Théophile Carniel, José Halloy, Jean-Michel Dalle.

**Methodology:** Théophile Carniel, José Halloy, Jean-Michel Dalle.

**Software:** Théophile Carniel.

**Supervision:** José Halloy, Jean-Michel Dalle.

**Visualization:** Théophile Carniel.

**Writing – original draft:** Théophile Carniel.

**Writing – review & editing:** Théophile Carniel, José Halloy, Jean-Michel Dalle.

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
