## [Decision Letter · Decision Letter 0]

7 Oct 2022

PONE-D-22-23540A novel clustering approach to bipartite investor-startup networks.PLOS ONE

Dear Dr. Theophile Carniel,

Thank you for submitting your manuscript to PLOS ONE. After careful consideration, we feel that it has merit but does not fully meet PLOS ONE’s publication criteria as it currently stands. Therefore, we invite you to submit a revised version of the manuscript that addresses the points raised during the review process.

We look forward to receiving your revised manuscript.

Kind regards,

Zilin Gao, Ph.D

Academic Editor

PLOS ONE

Journal Requirements:

2. Please note that PLOS ONE has specific guidelines on code sharing for submissions in which author-generated code underpins the findings in the manuscript. In these cases, all author-generated code must be made available without restrictions upon publication of the work. Please review our guidelines at https://journals.plos.org/plosone/s/materials-and-software-sharing#loc-sharing-code and ensure that your code is shared in a way that follows best practice and facilitates reproducibility and reuse. New software must comply with the Open Source Definition.

Reviewers' comments:

Reviewer's Responses to Questions

**Comments to the Author**

1. Is the manuscript technically sound, and do the data support the conclusions?

Reviewer #1: Yes

Reviewer #2: Yes

2. Has the statistical analysis been performed appropriately and rigorously? 

Reviewer #1: Yes

Reviewer #2: Yes

3. Have the authors made all data underlying the findings in their manuscript fully available?

Reviewer #1: Yes

Reviewer #2: Yes

4. Is the manuscript presented in an intelligible fashion and written in standard English?

Reviewer #1: Yes

Reviewer #2: Yes

5. Review Comments to the Author

Reviewer #1: The authors proposed a novel similarity-based clustering approach to venture capital investors that takes as input the bipartite graph of funding interactions between investors and startups and returns clusterings of investors built upon 5 characteristic dimensions. It is a topic of interest to the researchers in the relevant area, but the paper needs some improvement before acceptance for publication. My detailed comments are as follow.

Comments:

1. The motivation for the utilization of the proposed community detection algorithm based on the Hellinger-based similarity measure is unclear. Please specify this more clearly.

2. The Hellinger-based similarity measure plays a key role in the performance of community detection algorithm. There are many indices for the measures of distance and the similarity. such as the Manhattan distance measure and cosine similarity measure. Please explain the reasons for adopting the distance measure and similarity measure defined in Eq.(2) and Eq.(3), respectively.

3. English writing and grammar of the manuscript needs improvement. For instance, on page 7, line 105-106, the sentence “The log-binning was chosen as startup funding round amounts have been shown to follow a power-law distribution” is grammatically incorrect。

4. A number of labels of the figures are missing, for example, on Page 4, line 65, “see Fig. ??”. Please check this throughout the manuscript.

5. On the third last line of Page 4, “interacton”  “interaction”

Reviewer #2: Reports

This paper propose a similarity-based clustering approach to venture capital investors that takes as input the bipartite graph of funding interactions between investors and startups and returns clusterings of investors built upon 5 characteristic dimensions. Although this study is comprehensive and the results are interesting, I think that the paper needs a substantial revision before assessing whether it should be considered for publication. I give detailed comments below.

1. There are many clustering algorithms. Is your research an application of existing clustering algorithms to empirical evidence? Please clearly state your innovation, especially in terms of algorithm improvement or the advantages of your chosen clustering algorithm for the bipartite graph of funding interactions between investors and startups.

2. Please add the review of the literature about clustering algorithms of the bipartite graph and compare what advantages of the clustering method you choose in this paper.

3. When you built investor-startup network (Fig 1.), you said "As an investor can invest in a startup several times, multiple edges can connect two given nodes as represented by the arcs." Is there any difference between the arcs and edges？Do you treat them differently in the empirical？Why don't you use a weighted network to describe an investor invest in a startup several times？

4. Why you built upon these 5 characteristic dimensions? What are your reasons for considering this, and please explain them clearly.

5. In the conclusion you state only some of the phenomena you have observed through your study. You should state the practical implications of your study or how your conclusions will be applied in finance.

6. PLOS authors have the option to publish the peer review history of their article (what does this mean?). If published, this will include your full peer review and any attached files.

Reviewer #1: No

Reviewer #2: No

---

## [Author Response · Author response to Decision Letter 0]

24 Nov 2022

We thank the referees for carefully reading our article and for their constructive and helpful comments. Here are the comments and changes in the article.

1. The motivation for the utilization of the proposed community detection algorithm based on the Hellinger-based similarity measure is unclear. Please specify this more clearly.

- As the investors in our network are highly heterogeneous, we believe that a suitable methodology to cluster similar investors is necessary in order to study investment dynamics. The investor-startup network, however, is a complex bipartite multilayer multigraph, meaning that investors are multi-dimensional items defined along the various axes of the graph. Efficient clustering algorithms that take into account all the information present in the graph (corresponding

to edge and node attributes in the various layers), as far as we know, do not exist. We thus use domain knowledge from our team members in order to

build characteristic investor dimensions in order to create meaningful investors clusters. A paragraph has been added in the Introduction to address this point.

2. The Hellinger-based similarity measure plays a key role in the performance of community detection algorithm. There are many indices for the measures of distance and the similarity. such as the Manhattan distance measure and cosine similarity measure. Please explain the reasons for adopting the distance measure and similarity

measure defined in Eq.(2) and Eq.(3), respectively.

- Unlike for instance the Kullback-Leibler divergence, the Hellinger distance has all the properties of a distance measure. Here, we compute the distance be-

tween two low-dimensional probability distributions. Cosine distance (or similarity) is better suited for similarity measures in high-dimensional vector spaces. A number of other distance metrics exist, with the most frequently used being the Minkowski distance. The Minkowski distance, however, has been known to exhibit unintuitive behaviour in certain cases when comparing two probability distributions that the Hellinger distance does not show (see [1] for a more in- depth discussion). We thus decided to use the Hellinger distance for our study, as is often done when comparing probability vectors [2, 3]. We do note, however, that our results still qualitatively hold when using the Euclidean distance. A sentence has been added in the Hellinger distance and investor similarity subsection to justify this choice of distance.

3. English writing and grammar of the manuscript needs improvement. For instance, on page 7, line 105-106, the sentence “The log-binning was chosen as startup funding round amounts have been shown to follow a power-law distribution” is grammatically incorrect.

- This specific sentence has been reworded, and the manuscript has been reviewed.

4. A number of labels of the figures are missing, for example, on Page 4, line 65, “see Fig. ??”. Please check this throughout the manuscript.

- This was a LaTeX compilation problem that has now been corrected, our thanks to the reviewer.

5. On the third last line of Page 4, interacton → interaction

- This has been corrected, our thanks to the reviewer.

1. There are many clustering algorithms. Is your research an application of existing clustering algorithms to empirical evidence? Please clearly state your innovation, especially in terms of algorithm improvement or the advantages of your chosen clustering algorithm for the bipartite graph of funding interactions between investors and startups.

- As the investors in our network are highly heterogeneous, we believe that a suitable methodology to cluster similar investors is necessary in order to study investment dynamics. The investor-startup network, however, is a complex bipartite multilayer multigraph, meaning that investors are multi-dimensional items defined along the various axes of the graph. Efficient clustering algorithms that take into account all the information present in the graph (corresponding to edge and node attributes in the various layers), as far as we know, do not exist. We thus use domain knowledge from our team members in order to select characteristic investor dimensions in order to create meaningful investors clusters. A paragraph has been added in the Introduction to address this point.

2. Please add the review of the literature about clustering algorithms of the bipartite graph and compare what advantages of the clustering method you choose in this paper.

- Techniques to perform clustering on multi-view data exist [4, 5, 6], but are not able to deal with our specific constraints : our data is fundamentally bipartite, with each of the views containing different types of data (numerical vs. categorical vs. logarithmic) that is either node-based or edge-based. A generic method to deal with complex data of this format does not exist. Furthermore, our approach also has the added benefit of requiring a single parameter, which is the pruning threshold for the graph, and does not constraint the number of classes outputted. A paragraph has been added in the Introduction to address this point, and modifications have been made to the Objective section.

3. When you built investor-startup network (Fig 1.), you said ”As an investor can invest in a startup several times, multiple edges can connect two given nodes as represented by the arcs.” Is there any difference between the arcs and edges? Do you treat them differently in the empirical? Why don’t you use a weighted network to describe

an investor invest in a startup several times?

- The terms ”arcs” and ”edges” in this context describe the same thing, and so this has been corrected to only use a single term. A weighted network would not be suitable here as each individual edge contains important information such as the date of the investment, the funding amount of the investment and the stage of the investment; a multigraph is thus better-suited.

4. Why you built upon these 5 characteristic dimensions? What are your reasons for considering this, and please explain them clearly.

- These 5 characteristic dimensions were determined through the domain knowledge of some of our team members that have worked in entrepreneurial

ecosystems for over 20 years. Each of them represent a different investor facet, and correspond to characteristics related to the commonly agreed types of in-

vestors in entrepreneurship practice and research. This is detailed in the introduction of the Investor characterization subsection.

5. In the conclusion you state only some of the phenomena you have observed through your study. You should state the practical implications of your study or how your conclusions will be applied in finance.

- We provided additional information on the impact of our work, both in terms of practical implications for financial actors and in terms of research

implications for academics studying venture capital.

References

[1] Legendre P, Legendre L. Numerical ecology. Elsevier, 2012.

[2] Sohangir S, Wang D. Improved sqrt-cosine similarity measurement. Journal of Big Data, 4(1), 1-13.

[3] Zhu S, Lizhao L, Wang Y. Information retrieval using Hellinger distance and sqrt-cos similarity. 7th International Conference on Computer Science

& Education (ICCSE) (pp. 925-929). IEEE 2012.

[4] Wang H, Yang Y, Li B, Fujita H. A study of graph-based system for multi- view clustering. Knowledge-Based Systems, 2019, vol. 163, p. 1009-1019.

[5] Li L, He H. Bipartite graph based multi-view clustering. IEEE Transactions on Knowledge and Data Engineering, 2020

[6] Yang, Y, Wang H. Multi-view clustering: A survey. Big Data Mining and Analytics 1.2 (2018): 83-107.

---

## [Decision Letter · Decision Letter 1]

14 Dec 2022

A novel clustering approach to bipartite investor-startup networks.

PONE-D-22-23540R1

Dear Dr. Carniel,

We’re pleased to inform you that your manuscript has been judged scientifically suitable for publication and will be formally accepted for publication once it meets all outstanding technical requirements.

Kind regards,

Zilin Gao, Ph.D

Academic Editor

PLOS ONE

Additional Editor Comments (optional):

Reviewers' comments:

Reviewer's Responses to Questions

**Comments to the Author**

1. If the authors have adequately addressed your comments raised in a previous round of review and you feel that this manuscript is now acceptable for publication, you may indicate that here to bypass the “Comments to the Author” section, enter your conflict of interest statement in the “Confidential to Editor” section, and submit your "Accept" recommendation.

Reviewer #2: All comments have been addressed

2. Is the manuscript technically sound, and do the data support the conclusions?

Reviewer #2: Yes

3. Has the statistical analysis been performed appropriately and rigorously? 

Reviewer #2: Yes

4. Have the authors made all data underlying the findings in their manuscript fully available?

Reviewer #2: Yes

5. Is the manuscript presented in an intelligible fashion and written in standard English?

Reviewer #2: Yes

6. Review Comments to the Author

Reviewer #2: (No Response)

7. PLOS authors have the option to publish the peer review history of their article (what does this mean?). If published, this will include your full peer review and any attached files.

Reviewer #2: No

---

## [Editor Report · Acceptance letter]

22 Dec 2022

PONE-D-22-23540R1 

A novel clustering approach to bipartite investor-startup networks. 

Dear Dr. Carniel:

I'm pleased to inform you that your manuscript has been deemed suitable for publication in PLOS ONE. Congratulations! Your manuscript is now with our production department. 

Kind regards, 

on behalf of

Dr. Zilin Gao 

Academic Editor

PLOS ONE